# First Detection of the Adventive Egg Parasitoid of *Halyomorpha halys* (Stål) (Hemiptera: Pentatomidae) *Trissolcus mitsukurii* (Ashmead) (Hymenoptera: Scelionidae) in France

**DOI:** 10.3390/insects12090761

**Published:** 2021-08-24

**Authors:** Alexandre Bout, Francesco Tortorici, Rachid Hamidi, Sylvie Warot, Luciana Tavella, Maud Thomas

**Affiliations:** 1INRAE, UMR INRAE 1355 CNRS 7254, Institut Sophia Agrobiotech, Université Côté d’Azur, 400 route des Chappes, 06903 Sophia Antipolis, France; sylvie.warot@inrae.fr; 2Dipartimento di Scienze Agrarie, Forestali e Alimentari, University of Torino, Largo P. Braccini 2, 10095 Grugliasco, Italy; francesco.trt@gmail.com (F.T.); luciana.tavella@unito.it (L.T.); 3ANPN, Association Nationale des Producteurs de Noisettes, Lieu-dit Louberie, 47290 Cancon, France; rhamidi.anpn@unicoque.com; 4Unicoque, Noix et Noisettes de France, Lieu-dit de Louberie, 47290 Cancon, France; mthomas@unicoque.com

**Keywords:** biological control, tramp species, DNA barcoding, exotic species, brown marmorated stink bug

## Abstract

**Simple Summary:**

The invasive brown marmorated stink bug (BMSB), *Halyomorpha halys*, is a polyphagous species and a serious pest worldwide. Classical biological control (CBC), i.e., the introduction of a natural enemy from the native area of the pest, is considered as the preferential solution for BMSB regulation. Adventive populations of exotic egg parasitoids of BSMB, *Trissolcus japonicus* and *T. mitsukurii*, have been reported worldwide. With the aim to characterize the French diversity of egg parasitoids associated to various stink bugs including BMSB, field surveys were conducted between 2018 and 2020. Surprisingly, morphological and molecular analyses unambiguously revealed 12 specimens of *T. mitsukurii*, an Asian egg parasitoid of BSMB. Although its permanent establishment has to be confirmed, this first record in France could actually facilitate CBC initiatives, *T. mitsukurii* being considered with *T. japonicus* as the two most promising biological control agents against BMSB.

**Abstract:**

We report the first detection of *Trissolcus mitsukurii* in France. More than 1860 sentinel egg masses of *Halyomorpha halys* (BMSB) were exposed in the field during the 2018–2020 period, and 12 specimens of *T. mitsukurii* emerged from one egg mass. Their taxonomic identification was confirmed both by morphological and molecular analysis. *Trissolcus mitsukurii*, similar to *T. japonicus*, is an egg parasitoid of BMSB in its area of origin in Asia, and both species are considered to be candidates for a classical biological control strategy against BMSB. *Trissolcus mitsukurii* was previously recorded in Italy where it is well established and widespread, and this may be the source of the French population. Possible permanent establishment and dispersion of *T. mitsukurii* in France should be monitored with emphasis on its potential effect on BMSB populations.

## 1. Introduction

Numerous studies document the impact of biological invasions on biodiversity, human health, and economies [1]. Arthropods are among the most important invasive species [2,3,4,5], and the establishment of new agricultural and economic pests has increased drastically in the last few years [6,7], mainly due to the growth of trade and globalization.

Classical biological control (CBC) was developed to restore the ecosystem balance disrupted by newly arrived pests which are usually lacking natural enemies. Consequently, CBC can reduce the volume of chemical treatments required for their control. However, the development of CBC programmes to deliberately import and introduce exotic natural enemies requires regulatory approvals in many countries [8,9]. Prior to importation and release, researchers have to assess the risks and benefits of the potential introductions depending on each candidate species and concerned country. As explained by Mason et al. [10], the unintentional introductions of natural enemies are probably more common than expected. This phenomenon of “tramp” species was documented in Platygastroidea both for Platygastridae and Scelionidae [11,12].

In particular, the invasions of some emerging pests in Pentatomidae and Plataspidae (Hemiptera) in recent years have led to new CBC projects: *Halyomorpha halys* (Stål) [13,14], *Bagrada hilaris* (Burmeister) [14,15], and *Megacopta cribaria* (F.) [16,17]. Preliminary studies resulted in the detection of fortuitous populations of various exotic egg parasitoids in different countries, e.g., *Trissolcus japonicus* (Ashmead) in the USA [18,19], Canada [20], Switzerland, Italy [21,22], and Germany [23]; *Trissolcus mitsukurii* (Ashmead) in Italy [21,24] and Slovenia [25] against *H. halys*; *Trissolcus hyalinipennis* Rajmohana & Narendran in the USA [26] against *B. hilaris*; and *Paratelenomus saccharalis* (Dodd) in the USA [27] against *M. cribaria*.

France has not been spared by pests belonging to Pentatomidae: mainly, *Palomena prasina* (Linnaeus) (indigenous), the southern green stink bug, *Nezara viridula* (Linnaeus), and *H. halys*. *Halyomorpha halys* were firstly identified in France in April 2013, and the French Agency for Food, Environmental, and Occupational Health & Safety (ANSES) concluded that this invasive pest represents an important risk for many French crops [28]. In other countries, *H. halys* has become one of the major pests of apple orchards, causing more than 37 million dollars in losses in the Mid-Atlantic region of the USA in 2010 [29]. In France, since 2015, average bug damage on French hazelnut trees increased continuously from 0.2% in 2015 to up to 2% in 2018, with peaks that reached more than 14%. These data are in accordance with damage reported in hazelnut orchards in Italy and Georgia where *H. halys* is now established [30].

Among the natural enemies of *H. halys* in its native range, *T. japonicus* is regarded as the most promising candidate for CBC [31,32,33]. Numerous studies have been conducted to assess the parasitoid’s biosafety as part of the requirements to seek approval from the corresponding regulatory agencies for its legal introduction in the USA [34], Canada [20], New Zealand [35], and Europe [36]. However, in many cases, adventive populations of *T. japonicus* have also been reported [18,19,20,21,22,23,37,38,39]. *Trissolcus mitsukurii* was also recently found in northern Italy and western Slovenia [21,25]. This egg parasitoid is also known as a natural enemy of *H. halys* in its area of origin [21,32].

In France, neither *T. japonicus* nor *T. mitsukurii* has been detected until now. Therefore, since 2017, field monitoring was conducted to (i) clarify the French diversity of pentatomid egg parasitoids, (ii) identify the best potential candidate against the various stink bugs under consideration, and (iii) initiate preliminary studies in a context of CBC against *H. halys* prior to the exotic candidate introduction request.

## 2. Material and Methods

### 2.1. Field Surveys

Two main sampling methodologies were used to assess the diversity of the French egg parasitoids of stink bugs: exposure of sentinel egg masses and collection of field-laid egg masses (including *H. halys* and other Pentatomidae). The field surveys took place from 2017 to 2020 and covered different geographical areas (Table 1). Samples included field-collected egg masses and exposed sentinel egg masses as described below and egg masses occasionally provided by citizen scientist volunteers.

Sentinel egg masses

Regular exposure was performed in the Alpes Maritimes (southeastern France), close to the Italian border from 2018 to 2019, and in nut orchards in southwestern France in 2020. Occasional surveys were performed in other regions of France (Auvergne Rhône-Alpes, Bourgogne, Bretagne, Corse, Occitanie, and Pays de la Loire) during 2019.

Field-collected adults and nymphs of *H. halys* were transferred to the laboratory of INRAE Sophia Antipolis in 2016 and to the ANPN laboratory in 2020. They were reared in net cages (40 cm × 40 cm × 40 cm) (Bugdorm^®^, MegaView Science Co., Ltd., Taichung, Taiwan) containing fresh beans, complemented with fresh fruits, peanuts, or shelled hazelnuts. Mass rearing was performed at 25 ± 1 °C and RH 60 ± 10%, with an L:D of 16:8 h.

*Halyomorpha halys* egg masses were collected daily from the laboratory rearing (INRAE Sophia-Antipolis and ANPN) and directly exposed in the field or were frozen at −18 °C for at least 48 h. The use of fresh or frozen egg masses of *H. halys* depended on the preference of the owner of each experimental site.

Egg masses were glued to thin paper pieces (1 cm × 3 cm) and stapled to the undersides of the leaves of a wide variety of host plants, mostly woody trees (e.g., *Prunus* spp., *Acer* spp.) and shrubs (e.g., *Corylus avellana* L.). Sentinel egg masses were retrieved from the field three days after deployment.

Natural egg masses

Field-laid egg masses from *H. halys* and from other stink bug species were collected throughout France in 2019, and a collection in nut orchards in southwestern France was performed in 2020. During the surveys, egg masses of stink bug species were visually located on various host plants and collected.

Emergence of parasitoids

All egg masses, sentinel and field-laid, were reared at INRAE Sophia-Antipolis or ANPN until emergence of bug nymphs or parasitoid adults. Egg masses were individually placed in a glass tube (3 cm × 1 cm) together with a fine drop of honey and sealed with a cotton plug. Egg masses were kept under controlled conditions at 22 ± 1 °C and RH 60 ± 5%, with an L:D of 16:8 h. Each egg mass was checked daily for emergence, and the emerged parasitoid adults were stored in 96% ethanol at −18 °C.

### 2.2. Species Identification

DNA barcoding

DNA extraction was performed with 30 µL of buffer using the DNA extraction kit LUCIGEN (MA150E, QuickExtract™ DNA Extraction Solution, Middleton, WI, USA) (following company specifications. This method allows a non-destructive extraction of the DNA, so that the voucher (exoskeleton) remains intact for morphological identification.

PCR amplifications were performed on a portion of the Cytochrome Oxydase I subunit (COI) locus using the LCO-HCO primer: HCO2198 (5′-TAAA CTT CAG GGT GAC CAA AAA ATC A-3′), LCO1490(5′-GGTC AAC AAA TCA TAA AGA TAT TGG-3′) [40], allowing amplification of an approximately 600–700 bp portion of DNA on this locus. Product was sent to Beckman Coulter Genomics Genewiz (Leipzig, Germany, Essex, UK) for a double single way sequencing with HCO2198 primer. All residual DNA is archived at INRAE Sophia-Antipolis (France).

Correction, annotation, and alignment were performed manually using BioEdit Geneious R10 software. The comparison of nucleotide sequences with sequences available in the NCBI database (GenBank) was performed using Blastn [41] with standard settings. Analysis of sequence data was done with the MegaX software [42], using the neighbour joining (NJ) method [43], with Bootstrap values based on 100 replications. Nucleotide distances in NJ trees were estimated by the Kimura’s two-parameters method [44].

Morphological identification

Morphological identifications of *Trissolcus* specimens were performed using the specific taxonomic keys on *Trissolcus* [45,46]. Ethanol-stored and DNA-extracted specimens were mounted as recommended by Talamas et al. [46] and dried and glued on card points. Stereomicroscopes Leica M205C (INRAE) and Leitz large-field TS (DISAFA) were used for morphological diagnosis. Voucher specimens are located at INRAE UMR ISA in Sophia-Antipolis (southern France).

## 3. Results

Various species of native egg parasitoids belonging to Scelionidae (*Trissolcus* spp. and *Telenomus* spp.), Eupelmidae (*Anastatus bifasciatus* (Geoffroy)), and Encyrtidae (*Ooencyrtus* spp.) (unpublished data) emerged from egg masses of native Pentatomidae and *H. halys* collected during our surveys. All of the parasitoids collected are known to be parasitoids of stink bug eggs.

### Trissolcus mitsukurii

Among the fresh sentinel egg masses of *H. halys* exposed at Bergerac (France—Nouvelle-Aquitaine), one exposed on *Prunus laurocerasus* L. in mid-August 2020 was parasitized (Table 2) and produced 12 specimens of *T. mitsukurii*.

The specimens matched perfectly with the diagnosis provided in Talamas et al. [46] and Sabbatini-Peverieri et al. [21]. Females had an antennal clava of five clavomeres distinctly larger and darker than preceding flagellomeres. Other diagnostic features were the presence of an expanded orbital furrow at the intersection with the malar sulcus, the presence of a hyperoccipital carina posterior to the lateral ocelli and absent on the vertex, the absence of setae below the metapleural sulcus, rugose macrosculpture throughout the surface of the mesoscutellum, and the presence of a setal patch on the first laterotergite of the metasoma (Figure 1).

This identification was further confirmed by result of the partial sequencing of the COI gene from three specimens resulting in two available sequences but one unique COI haplotype of 600 bp. This haplotype does not have a stop codon in the predicted partial protein. These two sequences were deposited in GenBank (Table 2).

The best NCBI matches (100% identity) were obtained with the sequences gb |MK097190.1|, |MK097189.1|, |MN615594.1|, and |MN615595.1|, which correspond to *T. mitsukurii* specimens from Italy. The nucleotide sequences were then used to build a neighbour joining tree confirming the morphological identification (Figure 2). The geographical origins of the different *T. mitsukurii* sequences were indicated. The two sequences of *T. mitsukurii* from France were a retrieved cluster with the Italian specimens, well separated by a South Korean cluster and a cluster of Chinese and Japanese specimens.

## 4. Discussion and Conclusions

The sampling effort on French egg parasitoids of stink bugs allowed us to collect various species of Scelionidae, Eupelmidae, and Encyrtidae. This method is effective for the detection of *Trissolcus* species, including exotic ones [18,19,20,21,22,23,37,38,39].

Moreover, regarding the geographic range of our sampling, we consider it likely that we detected the first introduction of *T. mitsukurii* in France. A migration of the two exotic species *T. japonicus* and *T. mitsukurii* in southern France has been expected as (i) they are highly present and largely widespread in Italy and Switzerland, and (ii) the relative high density (but still lower than that observed in Italy) of *H. halys* provides an invasion opportunity for its parasitoids [47]. However, neither *T. japonicus* nor *T. mitsukurii* was collected in the Alpes-Maritimes close to Italy or close to Switzerland. Thus, the presence of *T. mitsukurii* in Bergerac, without populations along a gradient from East to West, appears to be the result of human activities, not natural dispersion. Moreover, the sampling point was not far from a plant nursery, which could be an indication that the parasitoids were transported with plant material. Regarding the COI clustering of the French population of *T. mitsukurii* with the Italian population, we consider that importation from Italy is more likely than direct importation from Asia.

This detection is insufficient to make a conclusion about the establishment of *T. mitsukurii* in France or its future dispersal. Nonetheless, the shared distribution of *T. mitsukurii* and *T. japonicus* in Asia, as well as their presence in Italy [47], could imply that their ecological preferences would be similar. According to Avila and Charles [48], French climatic conditions, as those of Europe, are favourable for widespread establishment of *T. japonicus* following the presence of its host, *H. halys*. The situation in Italy seems to indicate that the spread of *T. mitsukurii* is important, especially in comparison with *T. japonicus* which benefits from an artificial releasing programme [47]. This study also highlights the predicted spread of both *T. japonicus* and *T. mitsukurii* in all types of habitats where *H. halys* is present. Given that the climate of southern France is similar to northern Italy, we consider it likely that *T. mitsukurii* will become established. This is supported by the climate suitability model (CLIMEX Ecoclimatic Index) for *T. mitsukurii* performed by Yonow et al. [49]. Sampling in 2021 will be essential to determine if this is the case.

In a recent study aimed at defining the physiological host range of *T. mitsukurii* in Europe, *T. mitsukurii* displayed less host specialization in no-choice laboratory tests, similar to *T. japonicus* [50]. However, in choice tests, *T. mitsukurii* showed a similar host preference for more than one pentatomid species rather than its coevolved host *H. halys* [50], as opposed to *T. japonicus* [36]. Nonetheless, the parasitism of native species was rarely observed in the field [47].

At this point we cannot predict the ability of *T. mitsukurii* to control *H. halys* in France. Further studies are needed to determine if it is established and spreading in France and to assess non-target risks and potential side effects before complementary and augmentative releases are recommended.

## Figures and Tables

**Figure 1 insects-12-00761-f001:**
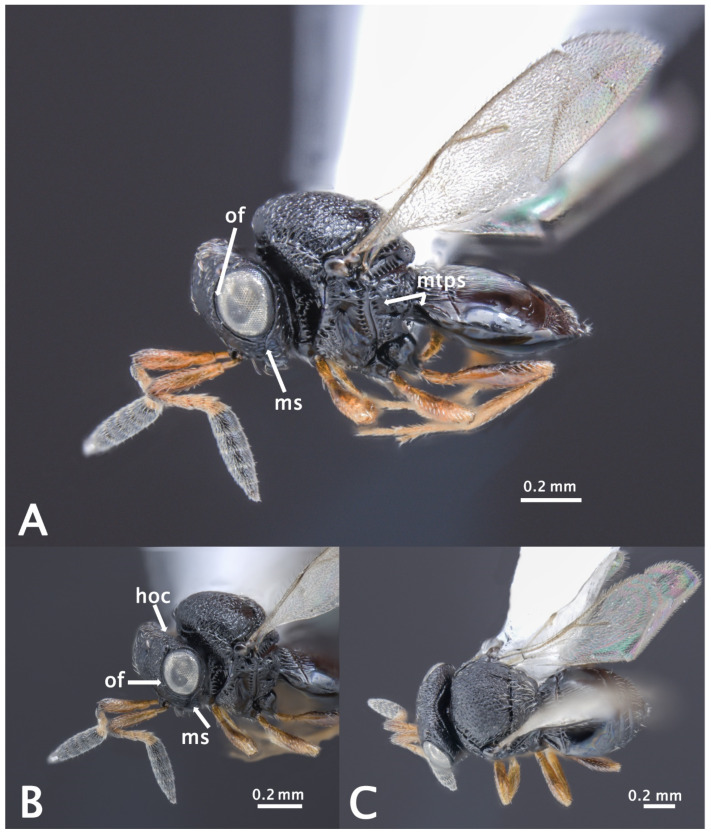
*Trissolcus mitsukurii*, voucher N° ISA_42315; (**A**) habitus in lateral view, metapleuron without setae below metapleural sulcus (mtps), (**B**) head with orbital furrow (of) and hyperoccipital carina (hoc), (**C**) habitus in dorsal view. Scale bars in millimetres.

**Figure 2 insects-12-00761-f002:**
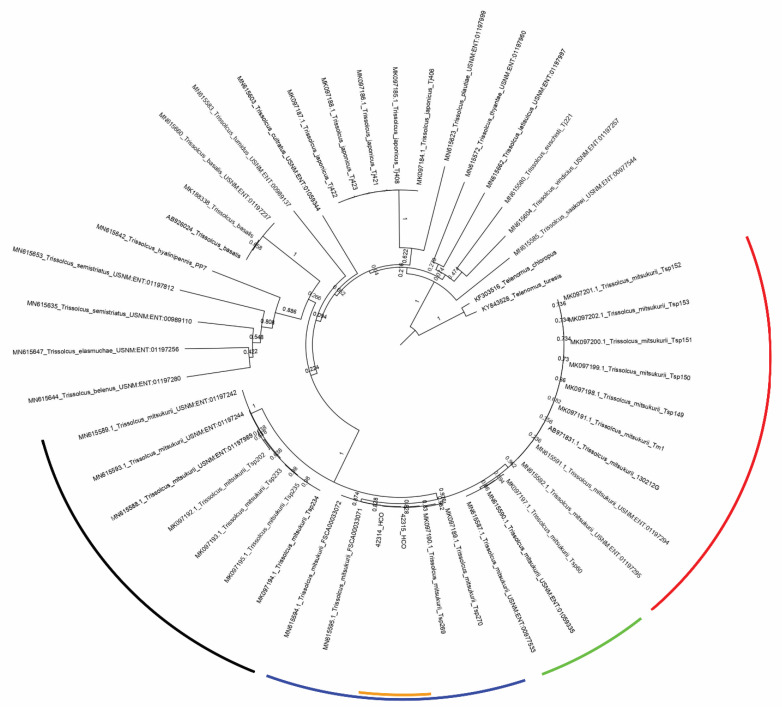
Molecular clustering of 48 sequences including the *Trissolcus mitsukurii* French individuals. All sequences of *T. mitsukurii* available in GenBank were used as some reference sequences of other *Trissolcus* species. *Telenomus turesis* Walker and *Te. chloropus* (Thomson) were used as out-group. Origins of *T. mitsukurii* are highlighted by colours: Japan in red, China in green, South Korea in black, Italy in blue, and French in orange.

**Table 1 insects-12-00761-t001:** Summary of the different main sampling areas.

Period	Area	Type of Sampling	Number of Egg Masses	Number of Parasitized Egg Masses	% of Parasitized Egg Masses
2018–2019	National-focus PACA	Sentinel-frozen	1020	223	21.82
		Sentinel-fresh	582	10	17.18
2020	National-focus PACA	Natural	129	60	46.51
May–September 2020	southwestern France	Sentinel-frozen	234	13	5.55
		Sentinel-fresh	15	2	13.33
May–September 2020	southwestern France	Natural	289	79	27.33

**Table 2 insects-12-00761-t002:** GenBank accession number and sample information for COI sequences presented in this study.

Species	Collection Code	Department Country	Year of Collection	GPS Coordinates (DMS)	Host Species	GenBank Accession Number
*Trissolcus mitsukurii*	42314_HCO	Lot, France	2020	44°50′8.21″, 0°29′48.534″	*Halyomorpha halys*	MZ343334
*Trissolcus mitsukurii*	42315_HCO	Lot, France	2020	44°50′8.21″, 0°29′48.534	*Halyomorpha halys*	MZ343335

## Data Availability

Data is contained within the article.

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
