# Peer review of "First Detection of the Adventive Egg Parasitoid of Halyomorpha halys (Stål) (Hemiptera: Pentatomidae) Trissolcus mitsukurii (Ashmead) (Hymenoptera: Scelionidae) in France"

_insects, 2021, doi:10.3390/insects12090761_

Round 1

Reviewer 1 Report

This is a very interesting study reporting on a multi-year survey of BMSB egg parasitoids conducted in France. The study itself it is quite simple, as well as its methodology, but results are quite relevant for a future CBC programme against BMSB in France, and therefore, of interest for the biocontrol community. I have not concerns about the methods, however, the study is missing very important information about the other parasitoid species recovered and their abundance. So, a table clearly showing all species found and their abundance must be presented as part of this work. After, if was a huge multi-year effort to do this work, so presenting only the data for T. mitsukurii is not enough.

The English language in this paper must be improved before this paper can be accepted for publication, as there are many sections very poorly written and unclear. I would recommend to seek help for a native English speaker to improve the manuscript. 

Please find all my specific comments, as well as some working suggestions to help to improve the writing of this manuscript.

Author Response

Many thanks for your valuable comments. 

Manuscript was revised for the English language according to the different reviewer. I add a version with active track changes to highlight correction.

Q1: How about the shield bug Pentatoma rufipes? Is this a problem in France? Powell 2020 reports this pest as an emerging pest of apples in northern Europe.

See: Powell 2020. The biology and control of an emerging shield bug pest, Pentatoma rufipes (L.) (Hemiptera: Pentatomidae). Agricultural and Forest Entomology, 22, 298–308. DOI: 10.1111/afe.12408

Answer: We keep in mind that there are emergences of Pentatoma rufipes in north European countries (Germany, UK). Currently, this species is not considered as an economical pest in France.

We reviewed for French agricultural technical journal the main and different issues with heteropteran on the various crop production (Streito et Bout, 2019) and we keep in contact with agricultural sectors through different research programs.

Streito, J.C. et Bout A. 2019. Punaises phytophages : quelle diversité sur nos cultures ? Ce deuxième volet présente les différentes familles d’hétéroptères d’intérêt agronomique, ainsi que la diversité des cultures concernées. Phytoma, N° 722, Bioagresseurs 53.

Q2: Regarding other parasitoids. You should actually present all this data in a table. It is of interest to see all number of parasitoids recovered and the corresponding species, so we have a better idea on existing parasitoids species complex and their abundance.

Answer:

First of all, the topic of this paper focus on the discovery of the first adventive population of Trissolcus mitsukurii in France. Adding other data could change the nature of this manuscript. Moreover, regarding French situation with BMSB outbreaks, legal rules for CBC programs, it is really important that this information should be published.

Then, integration of the data regarding others parasitoids will need more time, and change drastically the format of this manuscript, which is not our objective.

Finally, the valorization of these data, not presented in this manuscript, is planned in second stage.

Q3: Are these the sentinel egg masses deployed or recovered? You need to report on both.

Also, I see that below you mention that you used fresh and frozen egg masses as sentinel eggs. If this was the case, then you need to separate and report this data separately (i.e. fresh egg masses and frozen egg masses) because parasitism results may differ between fresh and frozen! - frozen eggs are usually less preferred by parasitoids, which is reflected in low parasitism

Answer:

Sentinel egg masses were deployed and recovered. We changed the tab 1 with separate data for frozen or not frozen sentinels egg masses. In our case frozen egg masses were more efficient in parasitoids results, as few of them (natives’ species) could not use “not frozen egg masses” of BMSB. Only Anastatus and Some Encyrtidae were recovered on not frozen sentinel egg masses of BMSB (or on natural laid egg masses of BMSB) ; exception of Trissolcus mitsukurii we reported here.

Q4 : From H. halys only? or also from other stink bug species? L108

I changed it in the text, to precise that these eggs come from BMSB and other species.

Q5: Or perhaps, sampling efforts in the other areas simply were not able to successfully recover the parasitoids! Remember that using sentinel egg masses for parasitoid surveys usually underestimates the actual abundance of parasitoids.

It could be a razon. However, we exposed frozen and not frozen egg sentinel egg masses in parallel of collection of natural laid egg masses, in various area, and specifically close to Italian border in Alpes maritimes, without recovering T. mitsukurii nor T. japonicus.

Reviewer 2 Report

Hi,

I think that this paper represents a huge amount of work, and it looks like it produced a very interesting result. Although the spread of T. mitsukurii is not a surprise, the merit of this paper is the extent to which sampling occurred and the early detection of the parasitoid. I have selected Major Revision because extensive editing of the language is needed, and there are some parts where clarification is needed. I have edited the attached version for English and I hope that it will help. There are a few technical issues- such as the captions for the figures, which are listed in the attached file. I look forward to seeing this paper published.

Author Response

Many thanks for your valuable comments. 

Manuscript was revised for the English language according to the different reviewer. I add a version with active track changes to highlight correction.

L59: Reviewer 2 suggestion is to replaced “bug damage” l59 by “BMSB damage”

Answer: We prefer to keep “bug damage” as at this step it was not possible to identify which heteropteran specie was responsible of damages. At least Palomena prasina, Gonocerus acuteangulatus and BMSB were present in high density where damages are encountered.

Even damages seem to increase in parallel of the increase of BMSB density, we can’t conclude in this relation.

Comment 1: What is meant by standardized exposure?

Answer: It mean regular exposition during sampling period. Changed in text.

Commented 2: These sites should be listed, either here or in the table.

Answer: other regions of sampling were listed in the text.

Commented 3: Does this mean throughout France?

Answer: Yes. I changed it in the text.

Commented 4: What do you mean by « systematically » ?

Answer: I mean that when we located egg masses we collected them. So, I deleted “systematically”.

Commented 5: I am not sure what this means.”double single way”

Answer: It correspond to the methods of sequencing from Genwise when we ask for a double sequencing of each DNA sample in forward and reverse sense in order to obtain a consensus sequence of the sample.

Comment 6: Neighbor joining trees are similarity-based, so it is probably best not to use the word phylogenetic.

Answer: I changed in the text according to the suggestion.

Commented 8: I think it should be mentioned that these are specimens from Italy. A 100% match to the Italian populations suggests that this is the source of the French population.

Answer: I added this information in the text.

Commented 9: What about the Slovenian specimens? Were they sequenced?

Answer: In Rot et al. 2021, the Slovenian specimens were not sequenced for DNA. They are considered in the publication, as a natural extension of the range of distribution of the Italian population.

Commented 10: I am not sure what this is stating.

“All sequences of T. mitsukurii, allowed in GenBank database, were used as some reference sequences of other Trissolcus species.”

Answer: deleted

Commented 12: The origin of the T. mitsukurii population is not well established.

Answer: I changed in the text to “the shared distribution in Asia” instead of “the shared origine”

Commented 13: What is this referring to?

Answer: I replaced “This example in Italy” by “Situation in Italy “which referred to the ref [47].

Commented 15: These letters do not match the images in the figure.

Answer: Caption was revised.

Commented 17: I suggest listing the color associations all in one place instead of at the beginning of the caption and at the end.

Answer: Color associations were listed in one place at the end of the caption.

Reviewer 3 Report

This is a straightforward study in which the presence of T. mitsukurii was documented in France for the first time. The manuscript needs some revision to conform with English style and language, and to that end, I have attached it with suggested changes in comment boxes. Otherwise, I have no additional comments.

Author Response

Many thanks for your valuable comments. 

Manuscript was revised for the English language according to the different reviewer. I add a version with active track changes to highlight correction.

L88: the owner or the researcher?

Answer: it was the owner. Each owner of sampling site could ask for frozen egg masses exposition rather fresh and viable egg masses to avoid a risk of contamination.

L91: After 3 days or within 3 days? Standard practice is after 3 days.

Answer: I changed for “after three 3 days”

L181: What do you mean by global? If truly global, i.e. from other countries, how did that "allow" you to conduct your work? Because of the identifications from global sampling?

Answer: we used “global” as it referred to surveys from different research programs used in this manuscript to clarify the total effort of egg parasitoids prospection. We deleted global to clarify.

Round 2

Reviewer 2 Report

Hi,

There are a few more corrections to the English in the attached version. Also, Figure 1 refers to setation of the metapleural sulcus, but as far as I know, this is not a character used in Trissolcus. Do you mean setation of the ventral metapleuron? I deleted the image in the attached version so that it would be small enough to upload.

Author Response

Dear Reviewer,

Thanks again for your comments.

I corrected english according to your recommendations

Other changes:

l21-22: I deleted "One of the nonexclusive explanations could be that this introduction is related to the Italian population where the parasitoid is widespread." as recommanded

l165 - Fig 1: caption were corrected regarding the comment.

Indeed, I changed "C) metapleural sulcus (mtps) without setae" by "metapleuron without setae below metapleural sulcus (mtps)"

Moreover I moved this last characteristic to the point "A)" and change the point "C)".

So final caption is : "Fig 1: Trissolcus mitsukurii, voucher n° ISA_42315; A) habitus in lateral view, metapleuron without setae below metapleural sulcus (mtps), B) head with orbital furrow (of) and hyperoccipital carina (hoc), C) habitus in dorsal view. Scale bars in millimeters."

In bibliography section:

I changed the initial ref n°50 as the preprint was published in Joural of Pest Science by

"50. Giovannini, L., Sabbatini-Peverieri, G., Marianelli, L., Rondoni, G., Conti, E. and Roversi, P.F. Physiological Host Range of Trissolcus Mitsukurii, A Candidate Biological Control Agent of Halyomorpha Halys in Europe. Journal of Pest Science, 2021, 102. https://doi.org/10.1007/s10340-021-01415-x"